# Impact of Nesting Mode, Diet, and Taxonomy in Structuring the Associated Microbial Communities of Amazonian Ants

Anaïs Chanson [1,2], Corrie S. Moreau [3] and Christophe Duplais [4,*]

1 Université de Guyane, UMR8172 Ecofog, Campus AgroParisTech, CNRS, Cirad, INRAE, Université des Antilles, 97304 Kourou, France
2 Department of Life Sciences, College of Science, University of Lincoln, Lincoln LN6 7TS, UK
3 Departments of Entomology and Ecology & Evolutionary Biology, Cornell University, Ithaca, NY 14850, USA
4 Department of Entomology, Cornell AgriTech, Cornell University, Geneva, NY 14456, USA
* Correspondence: c.duplais@cornell.edu; Tel.: +1-315-787-2429

**Abstract:** Studies of ant biodiversity are important to understand their group better, as well as to extend our knowledge on the evolution of their associated organisms. Host-associated microbial communities, and particularly bacterial communities, are shaped by different host factors such as habitat, diet, and phylogeny. Here, we studied the structures of bacterial and microbial eukaryote communities associated with Amazonian ants collected from two habitats: the rainforest and the city. We collected 38 ant species covering a large taxonomic range, and we used 16S rRNA and 18S rRNA amplicon sequencing to study the impact of the host's ecological and phylogenetic factors on their microbial communities. Our results show that (1) habitat does not structure ant microbial communities, (2) ant diet and nesting mode impact bacterial communities, while only nesting mode structures microbial eukaryote communities, and (3) microbial diversity is not correlated with host phylogeny, although several ant genera have conserved bacterial communities. As we continue to uncover the diversity and function of insect-associated microbes, this work explores how host ecology and evolutionary history shape ant microbial communities.

**Keywords:** Formicidae; insect; 16S rRNA; 18S rRNA; amplicon sequencing; metabarcoding

## 1. Introduction

One of the major goals in community ecology is to understand the diversity, maintenance, and consequences of biological interactions between different communities, especially in increasingly fragmented ecosystems [1]. Studies of ant biodiversity are not just relevant to an understanding of this globally dominant faunal group but also to an understanding of the evolution of associated organisms. Every organism has biological needs that can only be met under specific environmental conditions. Habitat filtering is one of the processes invoked to explain why, in a given habitat, only individual species possessing suitable traits for this specific habitat can persist [2,3]. Organisms rely on specific factors to survive and thrive. When organisms living in the same local environment depend on the same resources, they engage in competition for those resources, often leading to competitive exclusion [4]. In contrast, when certain organisms living in the same local environment do not depend on the same resources, they can coexist without competition, and this process is described as niche differentiation [3,5].

Microbial communities are ubiquitous and can be found in all types of environments, even the most extreme [6,7]. Environmental factors have been shown to have strong impacts on microbial richness and diversity in species-rich biomes [8–13]. Microbial communities are also often found in associations with diverse host organisms. In insects, symbiotic bacteria have shaped the evolution of their hosts and are key components in providing basic functions to their host [14,15]. Mutualistic bacteria are known to supplement nutritionally insufficient host diets [16–19]. They also help maintain and improve their host's health

and immune system [20,21] and participate in broadening their host's environmental tolerance [22–26]. On the other hand, parasitic bacteria can manipulate host reproduction [27] and limit host dispersion [28,29].

Compared to the study of insect-associated bacteria, scientific research on insect-associated microbial eukaryotes is not as widespread. Indeed, scientists tend to study insect-associated eukaryotes only when an insect shows outward signs of infection. Common signs of insect infection include behavioral and morphological changes such as climbing on the top of the canopy or a change in color and size. Many microbial eukaryotes including pathogenic fungi, nematodes, and protists have been identified as being associated with insects. Ascomycota fungi infect a wide range of insect hosts, and after killing an insect, often feed on its cadaver [30]. Basidiomycota and Chytriodiomycota fungi are known parasites of scale insects [31,32] and insect eggs [33,34], whereas Zygomycota occur on various insects without showing signs of host pathogenicity [35–38]. Regarding nematodes, several *Heterorhabditis* and *Steinerma* species have been reported as pathogens of coleopterans [39], dipterans [40], lepidopterans [41], and orthopterans [42]. Other nematodes also use insects as vectors but do not necessarily kill them. This is the case for several *Brugia* [43,44], *Dirofilaria* [45], *Onchocerca* [46], and *Wuchereria* [47] species, which use mosquitoes or midges as vectors to transmit diseases. Finally, among the stramenophiles, oomycota [48,49] are known to infect insects. Mutualistic associations between eukaryotes and insects are also widespread, especially between insects and fungi [50]. One of the most common examples is the cultivation of fungi for nutrition, which exists in several species of ants [51], beetles [52], and termites [53]. Mutualistic non-nutritional associations have also been demonstrated in numerous tripartite symbioses between the ascomycete (order: Chaetothyriales) fungi–plant–ant [54–56]. Another important aspect of mutualism between insects and fungi is the dispersion of fungi. Many insects have even evolved to have specific organs to carry fungal spores [57,58] or carry them in the gut [59]. Finally, insects also benefit from antimicrobial molecules produced by some fungi [60]. Mutualistic associations between insects and nematodes are rare, but a few examples are known [61], and some nematodes may have mutualistic associations with bacteria within insect hosts [62]. However, relying on observational methods limits the study of microbial eukaryote diversity in insects.

Among insects, ants represent a species-rich clade with different ecologies and provide many ecosystem services [63]. Symbiotic bacteria are thought to be partly responsible for their evolutionary success. For example, it has been hypothesized that symbiotic bacteria allow ants to dominate rainforest canopies [64,65]. Many studies have shown that symbiotic bacteria in ants differ depending on the ant diet, especially between herbivorous and carnivorous ants [66]. Predatory army ants possess specialized Firmicutes and Entomoplasmatales gut bacteria, which are common to all lineages of army ant with a symbiosis dating from the Cretaceous period and which are likely socially transmitted by trophallaxis or coprophagy [67,68]. The functional role of these bacteria in army ants is not yet known due to the lack of available symbiont genomes from army ants, but a nutritional role for these symbionts has been suggested [68]. Herbivorous turtle ants rely on symbiotic gut bacteria to recycle nitrogen into amino acids [69,70], while the Camponotoni tribe benefits from amino acid production by the obligate intracellular symbiont *Blochmannia* [19,71]. In addition, many studies have studied the bacterial communities of specific ant clades, such as *Daceton* [72], *Paraponera* [73], ponerine ants [74], *Pseudomyrmex* [75], *Solenopsis* [76], and spiny ants [77]. Several bacteria symbionts are associated across these multiple ant clades. For example, Acetobacterales, Entomoplasmatales, and Rhizobiales are bacterial orders commonly found in ant guts. Their role in ants is not fully understood, but Acetobacterales might be involved in larval immune function [78] and development [79], while Entomoplasmatales might be involved in chitin processing of insect prey [80], and Rhizobiales might be involved in protein degradation [81] and urea recycling pathways [82]. The bacteria symbiont *Wolbachia* is also found across several ant clades and may induce reproductive changes in hosts [83] and vitamin B supplementation [84], although in most cases, there

appear to be no positive or negative effects of *Wolbachia* infection for ants. Several microbial eukaryotes are also known to be associated with ants. The fungal class Sordariomycetes contains several species that infect ants, like the pathogen fungus infecting fungus-growing ants [85] or the fungus causing zombie ants [86]. Among the nematodes, many families have been shown to infect different ant species [87].

In this work, we aimed to study the factors structuring the microbial communities associated with Amazonian ants. We collected 38 ant species in French Guiana from a wide phylogenetic range spanning several ant subfamilies. We focused on the microbial communities associated in terms of both abundance and diversity using qPCR and 16S rRNA and 18S rRNA amplicon sequencing. Our sampling strategy had three objectives. First, a wide phylogenetic range was chosen to test correlations between microbial communities and the evolutionary history of their hosts. Second, we collected species possessing different ecological traits (diet and nesting mode) which were assessed for their correlations with bacterial communities. Third, among the 38 collected ant species, nine species were collected from both rainforest and city habitats to evaluate the impact of the environment on microbial communities. The results of this work contribute to our understanding of the different factors structuring the microbial communities associated with Neotropical ants.

## 2. Materials and Methods

### 2.1. Sample Collection

Samples were collected in March 2018 from two sites in French Guiana in the Nouragues Rainforest Reserve and the city of Cayenne (Figure S1). A total of 49 ant colonies were collected, representing 38 ant species from 18 genera spanning eight subfamilies. Several workers were collected from each colony with pincers and stored in small individual tubes containing ethanol. Then, tubes were kept at −20 °C until DNA extraction. Collected ant samples were identified up to the species level, when possible, in the field with a magnifying glass. Identifications were further refined in the lab with a binocular magnifier using the key to subfamilies of the Neotropical region from Baccaro et al. [88] The list of collected samples and different phylogenetic (subfamily, genera) and ecological factors (habitat, nesting mode, and diet) chosen is presented in Supplementary File S1. For 44 of the 49 sampled colonies, we also collected a sample of the nest material or foraging area. Nest samples were collected and stored in small sterile bags. Then, bags were kept at −20 °C until DNA extraction. Vouchers for all samples were deposited in the Cornell University Insect Collection (Ithaca, New York, NY, USA).

### 2.2. DNA Extractions

DNA extractions from single ants and nest samples were performed using the DNeasy PowerSoil Kit (Qiagen, Germantown, MD, USA) following the manufacturer's protocol with a few modifications before the first step. For the DNA extractions, a single worker from each collected nest was used, and between 0.1 and 0.25 g of nest material was used. First, ant samples and nest samples were introduced into sterile 1.5 mL Eppendorf tubes. Then, 1 mL of liquid nitrogen was added to each Eppendorf tube, and each sample was immediately manually crushed in the tube with a sterile micropestel. To avoid any contamination between samples during this step, a new sterile micropestel was used for each sample tube. The crushed samples were then transferred to tubes with 500 µL PowerSoil bead solution. Following the procedure suggested by Rubin et al. [89], 60 µL of solution C1 and 100 µg of proteinase K were added to each PowerSoil tube, and tubes were incubated at 56 °C overnight. The extraction then proceeded following the DNeasy PowerSoil Kit protocol. The same protocol was followed with 4 blank tubes containing no ant or nest sample, and these served as negative controls. Filtered pipette tips and sterile techniques were used in every step to avoid contamination [90]. All DNA extractions were quantified via Qubit to verify the success of the DNA extraction. The Qubit quantification was performed with the High Sensitivity Assay Kit (Life Technologies Corp., Carlsbad, CA, USA).

### 2.3. DNA Amplification

Amplification of the V4 region of bacterial 16S rRNA for the ant and nest dataset and V1–V2 region of 18S rRNA of the ant dataset as well as Miseq sequencing of each DNA dataset were performed by the Argonne National Laboratory (Lemont, IL, USA). Negative controls were also processed following the same protocol. Following the protocol suggested in the Earth Microbiome Project (EMP) (http://www.earthmicrobiome.org/protocols-and-standards/16S/, accessed on 9 December 2019), amplifications for the V4 region of 16S rRNA were performed using 515F (5′-GTGCCAGCMGCCGCGGTAA) and 806R (5′-GGACTACHVGGGTWTCTAAT) primers, as described by Caporaso et al. [91], and amplifications for the V1–V2 region of 18S rRNA were performed using F04 (5′-GCTTGTCTCAAAGATTAAGCC) and R22 (5′-GCCTGCTGCCTTCCTTGGA) primers, as described by Creer et al. [92]. Each PCR reaction contained 12 µL of DNA-free PCR water, 10 µL of 5 Prime HotMasterMix 1X, 1 µL of 5 mM forward primer, 1 µL of 5 mM Golay barcode tagged reverse primer, and 1 µL of extracted DNA. The amplification conditions were as follows: 94 °C for 3 min, with 35 cycles at 94 °C for 45 s, 50 °C for 60 s, and 72 °C for 90 s, and a final cycle of 10 min at 72 °C. Each PCR reaction was performed in triplicate. Electrophoresis with 1% agarose gel was performed to confirm the efficiency of the amplification. Amplification samples from ants and their nests were pooled separately with each pool containing a 100 µL samplr, and these were cleaned using the QIAquick PCR Purification Kit (Qiagen, USA) following the manufacturer's instructions. The molarity of the pool was determined and diluted down to 4 nM, denatured, and then diluted to a final concentration of 6.75 pM with 10% PhiX for sequencing. Three separate runs (one for the 16S rRNA ant dataset, one for the 16S rRNA nest dataset and one for the 18S rRNA ant dataset) were performed with the MiSeq Illumina V3 Reagent Kit 600 Cycles (300 × 300) using the custom sequencing primers and procedures described in the Supplementary Methods by Caporaso et al. [91] for 16S rRNA and Creer et al. [92] for 18S rRNA.

### 2.4. Bacterial Quantification

Bacterial quantification of each sample was performed through qPCR quantification (Thermo Fisher Scientific, Waltham, MA, USA) on real-time CFX Connect equipment (Bio-Rad). A SYBRAdvanced 2X (Bio-Rad) SYBR green supermix and 2 µL of extracted DNA were used to verify the total amount of bacteria present in each sample. Amplification of the V4 region of 16S rRNA in the qPCR was performed using 515F and 806R primers, as described by Caporaso et al. [91], following the protocol suggested in the Earth Microbiome Project (EMP) (http://www.earthmicrobiome.org/protocols-and-standards/16S/, accessed on 9 December 2019). Each qPCR reaction was performed in triplicate. Standard curves were generated from serial dilutions of linearized plasmids containing *E. coli* 16S rRNA inserts, following the same parameters of Rubin et al. [89]. All triplicate qPCRs values were satisfactory and had $R^2$ values from 70% to 100%. The mean triplicate value for each sample was used in the analysis. Negative controls were also analyzed following the same protocol.

### 2.5. Bacterial and Microbial Eukaryote Diversity

Demultiplexing of sequences and taxonomic assignments was performed separately for the ant and nest sequences but following the same protocol. Demultiplexed sequences were analyzed using Qiime2-2019.1 [93] with the plugin demux (https://github.com/qiime2/q2-demux, accessed on 22 August 2020). Sequence quality control and feature table construction were performed through the dada2 plugin [94]. Taxonomic assignment was conducted with the SILVA_132_QIIME database [95], and the ASVs (amplicon sequence variants) were selected with 99% identity. To generate the taxonomy table, paired-end sequence reads were trimmed in the V4 region of 16S rRNA with the 515F/806R primers and in the V1–V2 region of 18S rRNA with the F04/R22 primers. Thereby, our own classifier was created using the "feature-classifier fit-classifier-naive-bayes" command. Once the

classifier was obtained, the reads (rep-seqs) were classified by taxon using the "feature-classifier classify-sklearn" command [96].

The filtration of contaminants (blank samples) in the datasets was performed with the Decontam package [97] of R software version 4.02 [98]. In this package, the prevalence method was used to remove the contaminant sequences from our samples. The decontaminated datasets were inserted back into Qiime2 [93] to filter all mitochondria and chloroplast sequences from the datasets. Hymenopteran sequences were also excluded from the 18S rRNA table so that sequences coming from the ant host did not appear in our analyses. The alignment was performed using the align-to-tree-mafft-fasttree command [99] to reconstruct the microbial phylogenies.

The alpha and beta diversity analyses were performed by using the "qiime diversity core-metrics-phylogenetic" command. Beta diversities were visualized using the visualization interface https://view.qiime2.org/, accessed on 22 August 2020. Alpha diversity metrics computed were the Shannon index, the Pielou's evenness index, the Faith's phylogenic diversity, and the number of ASVs. Beta diversity metrics computed were the Jaccard similarity index, Bray–Curtis dissimilarity, unweighted unifrac distance, and weighted unnormalized unifrac distance. The Jaccard similarity index gauges the similarity and diversity without accounting for the abundance [100]. The Bray–Curtis dissimilarity is a measure of the overabundant taxa [101]. The unweighted unifrac distance measures the unique branch length [102], and the weighted unnormalized unifrac distance estimates the abundance but does not correct for different evolutionary rates between taxa [103].

### 2.6. Statistical Analysis

Statistical analyses of bacterial quantification were performed using one-way ANOVAs (Analysis of Variance) with PAST software version 4.02 [104]. Statistical analyses of alpha diversity and beta diversity were performed on Qiime2-2019.1 [93] using the "diversity alpha-group-significance" command with the pairwise Kruskal–Wallis methods, and the "diversity beta-group-significance" command with the pairwise PERMANOVA method and 999 permutations. The bacterial quantification and alpha diversity results were visualized using PCoAs (Principal Coordinates Analysis) generated with the R packages "ggpubr" version 0.3.0 [105] and "PMCMR" version 4.3 [106]. The beta diversity results were visualized using PCoAs with the R packages "ggplot2" version 3.3.0 [107] and "ggfortify" version 0.4.10 [108]. The correlation between ant evolutionary history and microbial composition was determined using Mantel tests using the R package phytools [109]. The contribution of each ASV to each sample at the order level (for the 16S rRNA datasets) and at the phyla level (for the 18S rRNA dataset) was determined using the "qiime taxa barplot" and "qiime taxa collapse" commands on Qiime2-2019.1 [93]. SIMPER (Similarity Percentage) analyses were performed using the PAST software version 4.02 [104] and visualized using boxplots with the R packages "ggpubr" version 0.3.0 [105] and "PMCMR" version 4.3 [106].

## 3. Results

### 3.1. 16S rRNA Assessing Sequencing Quality

A total of 49 ant samples and 44 nest samples were sequenced with four control samples, two for each sample type (Supplementary File S1). For the ant samples, a total of 1,879,666 reads were sequenced. After rarefaction at a sampling depth of 4500 reads, three samples were removed due to a low number of reads, resulting in a total of 5908 ASVs recovered from the 46 ant samples, ranging from 7548 to 80,226 reads, with a mean frequency of 40,842 reads per sample. The rarefaction curve shows that sequences from every sample reached a plateau, indicating that most of the bacterial diversity was recovered (Figure S2A). Three samples (CSM3695a, CSM3695b and PJF10) were excluded from the dataset because they did not reach the minimum sampling depth of 4500 reads. Nest samples were rarefied to 9000 reads (Figure S2B), and in total, 19,934 ASVs were obtained from 2,350,607 reads sequenced in the 44 nest samples, ranging from 9130 to 76,765 reads, with a mean frequency of 53,442 reads per sample.

### 3.2. 16S rRNA Alpha Diversity

First, we tested for dissimilarities in bacterial alpha diversity across ant habitat, nesting mode, diet and subfamily. There was no differences in bacterial alpha diversity between rainforest ants and city ants (Figure S3; *Shannon: H = 0.017, p-value = 0.895; Pielou: H = 0.002, p-value = 0.965; Faith: H = 0.213, p-value = 0.644; ASV: H = 0.148, p-value = 0.700*), However, there were differences in bacterial alpha diversity across nesting modes (Figure S4; *Shannon: H = 15.563, p-value = 4.17E-04; Pielou: H = 13.742, p-value = 0.001; Faith: H = 12.159, p-value = 0.002; ASV: H = 16.350, p-value = 2.817E-04*), diets (Figure S5; *Shannon: H = 4.316, p-value = 0.116; Pielou: H = 6.431, p-value = 0.040; Faith: H = 0.424, p-value = 0.809; ASV: H = 0.456, p-value = 0.796*) and subfamilies (Figure S6; *Shannon: H = 18.096, p-value = 0.012; Pielou: H = 17.280, p-value = 0.016; Faith: H = 12.113, p-value = 0.097; ASV: H = 17.834, p-value = 0.013*).

The SIMPER (Similarity Percentage) analysis was performed to determine the most abundant bacterial orders in the ant samples. The taxa bar plot of the bacterial relative abundance exhibited very diverse patterns across ant genera (Figure 1), and across the habitat, nesting mode, and diet (Supplementary File S2). The 15 most abundant bacterial orders found in the host samples were, in order, Rickettsiales, Rhizobiales, Enterobacteriales, Acetobacterales, Lactobacillales, Burkholderiales, Xanthomonadales, Erysipelotrichales, Flavobacteriales, Pseudomonadales, Corynebacteriales, Entomoplasmatales, Opitutales, Sphingomonadales, and Micrococcales (Supplementary File S3).

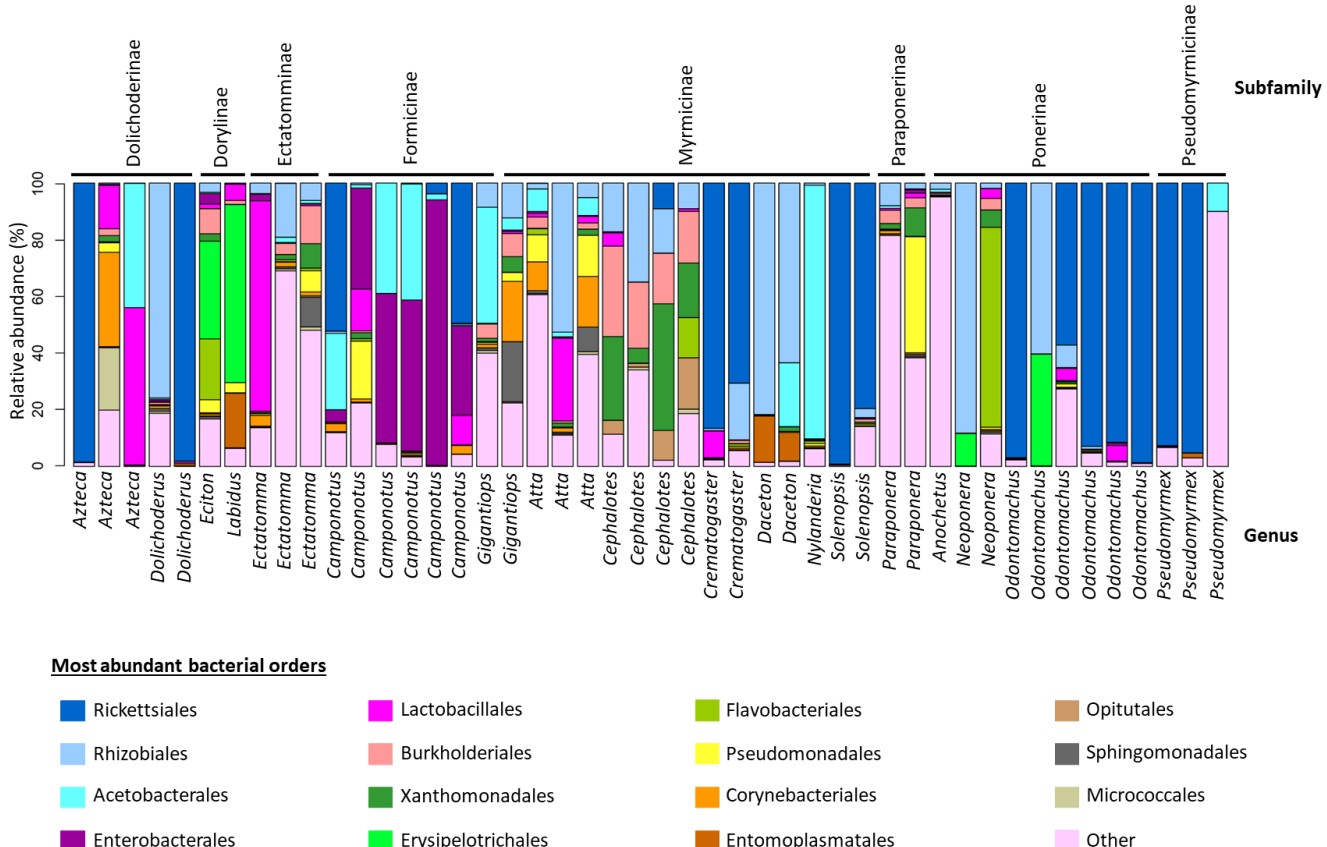

**Figure 1.** Taxa bar plot based on the relative abundance classified by ant subfamily and genus. Relative abundance bars are colored by bacterial order percentage. Only the 15 most abundant bacterial order associated with Amazonian ants are listed. The less abundant bacterial orders are grouped under the term "Other".

Rickettsiales were present (at least 10% of relative abundance) in every habitat, diet, and nesting mode tested (Supplementary File S2). Rhizobiales were also very common,

being found across carnivorous and herbivorous ants (carnivorous: 21.38%; herbivorous: 23.63%) as well as in ants nesting in trees (21.36%) and in the ground (10.25%) and in rainforest ants (16.87%). Acetobacterales and Enterobacteriales were present in omnivorous ants (respectively, 13.76% and 12.71%) and ants nesting in rotten logs (respectively, 24.3% and 28.69%). Acetobacterales were also found in city ants (10.93%). Burkholderiales and Xanthomonadales were common in herbivorous ants (respectively, 11.00% and 10.98%), while Erysipelotrichales were only found in carnivorous ants (9.29%).

The bacterial relative abundance showed different patterns across ant genera but was relatively consistent within the same genus. This was especially true for *Camponotus*, *Cephalotes*, *Crematogaster*, *Daceton*, *Solenopsis*, and *Odontomachus* (Figure 1). *Crematogaster* and *Odontomachus* were found to have very conserved bacterial compositions, consisting primarily of Rickettsiales (respectively 78.58% and 72.87%) and Rhizobiales (respectively 10.85% and 11.60%). Rickettsiales was also found to be the most abundant bacterial order in *Solenopsis* (59.59%). *Camponotus* were dominated by Enterobacteriales (45.19%), but also contained Acetobacterales (18.47%) and Rickettsiales (17.60%). *Daceton* were dominated by Rhizobiales (72.49%) but also had Entomoplasmatales (13.20%) and Acetobacterales (11.31%). *Cephalotes* were associated with Xanthomonadales (24.65%), Burkholderiales (22.71%), and Rhizobiales (19.19%). In addition, some of the most abundant bacterial orders were only found in a few ant genera. Erysipelotrichales (Firmicutes) were only found in *Eciton* (34.47%) and *Labidus* (62.80%). *Eciton* also contained Flavobacteriales (21.37%), while *Labidus* also possessed Entomoplasmatales (19.24%).

*3.3. 16S rRNA Beta Diversity Analysis*

The bacterial diversity of the ants was compared with the bacterial diversity of their nest samples, and both the Jaccard and unUniFrac distances showed statistical differences (Figure S7A,B; *Jaccard: pseudo-F = 2.967, p-value = 0.001; unUniFrac: pseudo-F = 8.162, p-value = 0.001*). The forest/city dataset showed the same statistical differences (*Jaccard: pseudo-F = 1.919, p-value = 0.001; unUniFrac: pseudo-F = 3.535, p-value = 0.001*). In the rest of the manuscript, we only focused on the analyses of the ant samples.

Then, we tested for dissimilarities in bacterial diversity across ant habitat, diet, nesting mode, and taxonomy. All statistics for each beta diversity metric are reported in Supplementary File S4. The bacterial qPCR quantification analysis revealed no statistical differences in bacterial quantification associated with ants across ant taxonomy, diet, nesting mode, and habitat (Figure S8; *df = 46; Habitat: t-test, t = 1.438, p-value = 0.157; Diet: ANOVA, F = 0.193, p-value = 0.956; Nesting mode: ANOVA, F = 0.128, p-value = 0.957; Subfamily: ANOVA, F = 0.400, p-value = 0.813*). There were no statistical differences either in the forest/city dataset across ant habitat, diet, nesting mode, or taxonomy (*df = 20; Habitat: t-test, t = 1.345, p-value = 0.194; Diet: ANOVA, F = 1.761, p-value = 0.106; Nesting mode: ANOVA, F = 1.823, p-value = 0.141; Subfamily: ANOVA, F = 0.331, p-value = 0.941*).

The bacterial beta diversity did not differ between rainforest ants and city ants (Figure 2A,B, Supplementary File S4; *Bray-Curtis: pseudo-F = 0.920, p-value = 0.619; wUniFrac: pseudo-F = 0.945, p-value = 0.502*); however, the nesting modes ground nesting, rotten log, and tree nesting all showed statistically different bacterial diversity results (Figure 2C,D, Supplementary File S4; *Bray-Curtis: pseudo-F = 1.473, p-value = 0.005; wUniFrac: pseudo-F = 2.217, p-value = 0.002*). Overall, the diet was a strong factor in differential bacterial diversity (Figure 2E,F; *Bray-Curtis: pseudo-F = 1.361, p-value = 0.018; wUniFrac: pseudo-F = 1.675, p-value = 0.029*). Carnivorous ants were shown to have different bacterial communities than herbivorous ants, which had different bacterial communities to omnivorous ants, but no difference was found between carnivorous and omnivorous ants (Supplementary File S4). Concerning ant taxonomy, the results were statistically different across subfamilies (Figure 3A,B; *Bray-Curtis: pseudo-F = 1.368, p-value = 0.018; wUniFrac: pseudo-F = 1.715, p-value = 0.025*). More specifically Dorylinae, Formicinae, Myrmicinae, and Ponerinae had different bacterial compositions to other subfamilies (Supplementary File S4). To test whether these dissimilarities are due to the ants' evolu-

tionary histories, we performed Mantel tests comparing the bacterial diversity with the host phylogeny [110]. However, the Mantel tests showed no correlation between bacterial diversity and ant phylogeny for both the Bray–Curtis and wUniFrac distances (Figure 3C; *Bray–Curtis: F = 1.473, p-value = 0.516; wUniFrac: F = 2.217, p-value = 0.498*). The forest and city datasets gave the same statistical results for every factor tested (Supplementary File S4).

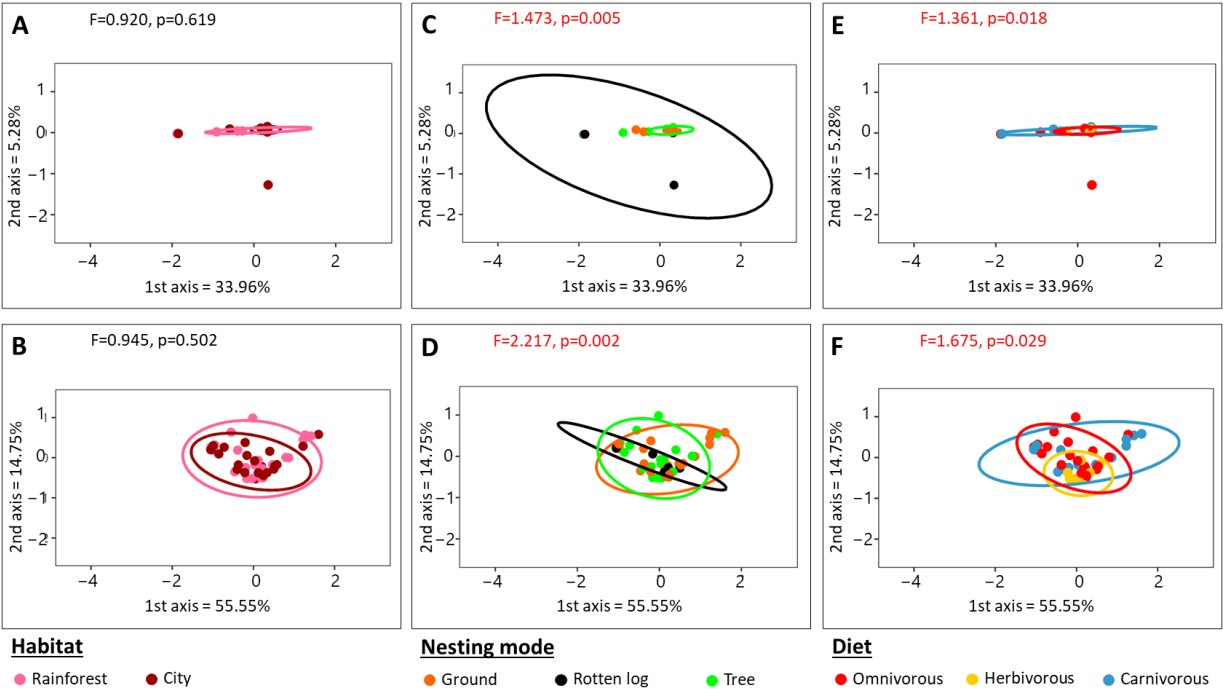

**Figure 2.** Differences in bacterial communities visualized with PCoAs across ant habitats (**A**,**B**), nesting modes (**C**,**D**), and diets (**E**,**F**). The distances used were the Bray–Curtis (**A**,**C**,**E**) and wUniFrac (**B**,**D**,**F**). The statistical *p*-values were obtained with pairwise PERMANOVAs. The circles represent confidence intervals of 0.95.

### 3.4. 16S rRNA ASV Similarity Percentage Analysis

Next, we used a SIMPER analysis to identify the most abundant bacterial orders responsible for the dissimilarities observed previously. This analysis identified 15 bacterial orders responsible for the differences. All statistics for the SIMPER analyses are presented in Supplementary File S5. Erysipelotrichales were more abundant in carnivorous ants (Figure S9A), while Rhizobiales were more abundant in herbivorous ants (Figure S9C). However, both were also more abundant in ground nesting ants than in rotten log nesting ants (Figure S9B,D). Lactobacillales were more abundant in ground nesting ants compared to ants nesting in rotten logs, while Rickettsiales had a higher abundance in rotten log nesting ants than in tree nesting ants (Figure S9F,H). Entomoplasmatales were more abundant in carnivorous ants than in herbivorous ants (Figure S9E). Finally, Acetobacterales had a greater abundance in city ants than in rainforest ants (Figure S9G). However, there was no statistical difference in any bacterial order between ant subfamilies. Pairwise Kruskal–Wallis comparisons for all the discussed bacterial orders are presented in Supplementary File S6.

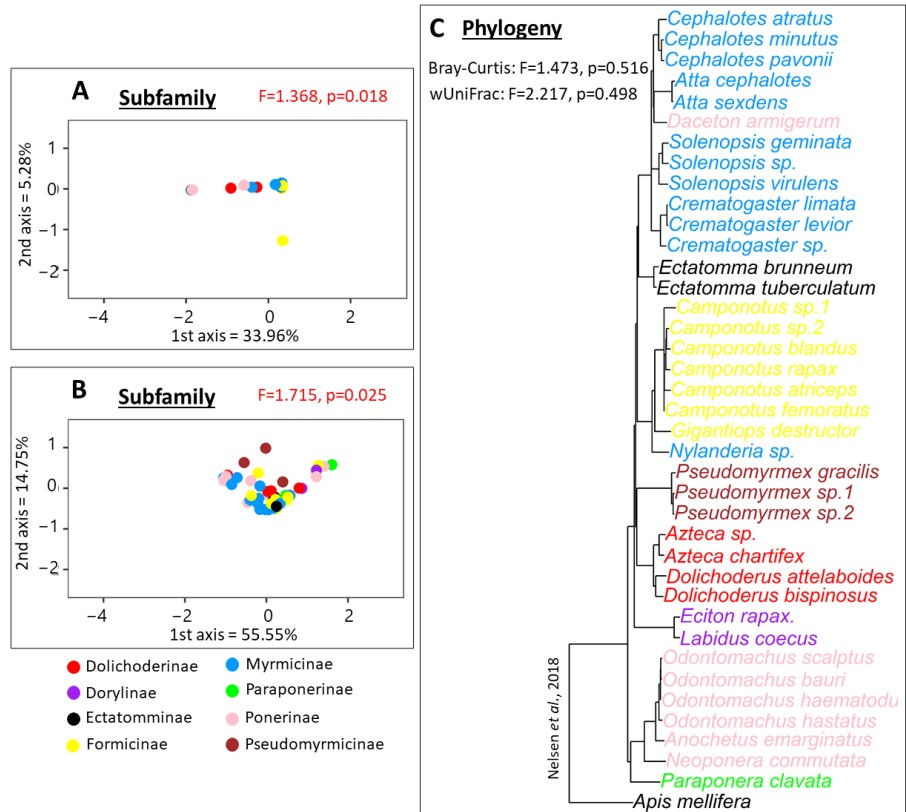

**Figure 3.** Differences in bacterial communities visualized with PCoAs across ant subfamily (**A**,**B**) and phylogeny (**C**). The distances used were the Bray–Curtis (**A**,**C**) and wUniFrac (**B**,**C**). The statistical *p*-values were obtained with pairwise PERMANOVAs.

### 3.5. 18S rRNA Assessing Sequencing Quality

We assessed the eukaryote diversity associated with the same 49 Amazonian ants used for the bacterial diversity analysis (Supplementary File S1). In the raw dataset, there was a total of 21,719 reads. After rarefaction of the samples to 199 reads, 33 samples were excluded from the dataset because they did not reach the minimum sampling depth (Figure S10). In total, from the 16 remaining samples, we obtained 118 ASVs from 20,185 reads ranging from 199 to 7815 reads with a mean frequency of 1261 reads per sample.

### 3.6. 18S rRNA Alpha and Beta Diversity Analysis

Alpha diversity analyses were performed using Qiime2 software to test for dissimilarities in eukaryote diversity in our dataset. There was no differences in microbial eukaryote alpha diversity between rainforest ants and city ants (Figure S11; *Shannon: H = 0.540, p-value = 0.462; Pielou: H = 0.011, p-value = 0.916; Faith: H = 1.103, p-value = 0.294; ASV: H = 1.773, p-value = 0.183*), nesting mode (Figure S12; *Shannon: H = 4.352, p-value = 0.360; Pielou: H = 5.593, p-value = 0.200; Faith: H = 3.643, p-value = 0.456; ASV: H = 2.581, p-value = 0.630*), diet (Figure S13; *Shannon: H = 3.824, p-value = 0.148; Pielou: H = 1.283, p-value = 0.526; Faith: H = 3.107, p-value = 0.212; ASV: H = 4.634, p-value = 0.099*) or subfamily (Figure S14; *Shannon: H = 8.184, p-value = 0.017; Pielou: H = 4.634, p-value = 0.099; DOI, p-value = 0.212; ASV: H = 1.283, p-value = 0.526*).

Beta diversity analyses were performed using Qiime2 software to test for dissimilarities in eukaryote diversity in our dataset. Statistics for the two beta diversity metrics tested (Bray–Curtis distance and wUnifrac distance) are presented in Supplementary File S8.

For the two metrics tested, there were no differences between ant habitats (Figure 4A,B; Bray–Curtis: pseudo-F = 1.042, *p*-value = 0.320; wUnifrac: pseudo-F = 1.295, *p*-value = 0.199), diets (Figure 4E,F; Bray–Curtis: pseudo-F = 1.041, *p*-value = 0.281;

wUnifrac: pseudo-F = 1.113, *p*-value = 0.313), or subfamilies (Figure 5A,B; Bray–Curtis: pseudo-F = 1.043, *p*-value = 0.245; wUnifrac: pseudo-F = 0.718, *p*-value = 0.652). There was also no correlation between the eukaryote diversity and ant evolutionary history, as shown by the Mantel test (Figure 5C; Bray–Curtis: F = 0.074, *p*-value = 0.227; wUnifrac: F = 0.064, *p*-value = 0.603). Ground nesting ants showed statistically different results to tree nesting ants (Figure 4C,D; Bray–Curtis: pseudo-F = 1.121, *p*-value = 0.048; wUnifrac: pseudo-F = 3.143, *p*-value = 0.029). Omnivorous ants also showed statistically different results to carnivorous and herbivorous ants for the Bray–Curtis distance (Figure 4E; F = 1.041, *p*-value = 0.033); however, these differences did not appear for the wUnifrac distance (Figure 4F; F = 1.113, *p*-value = 0.313).

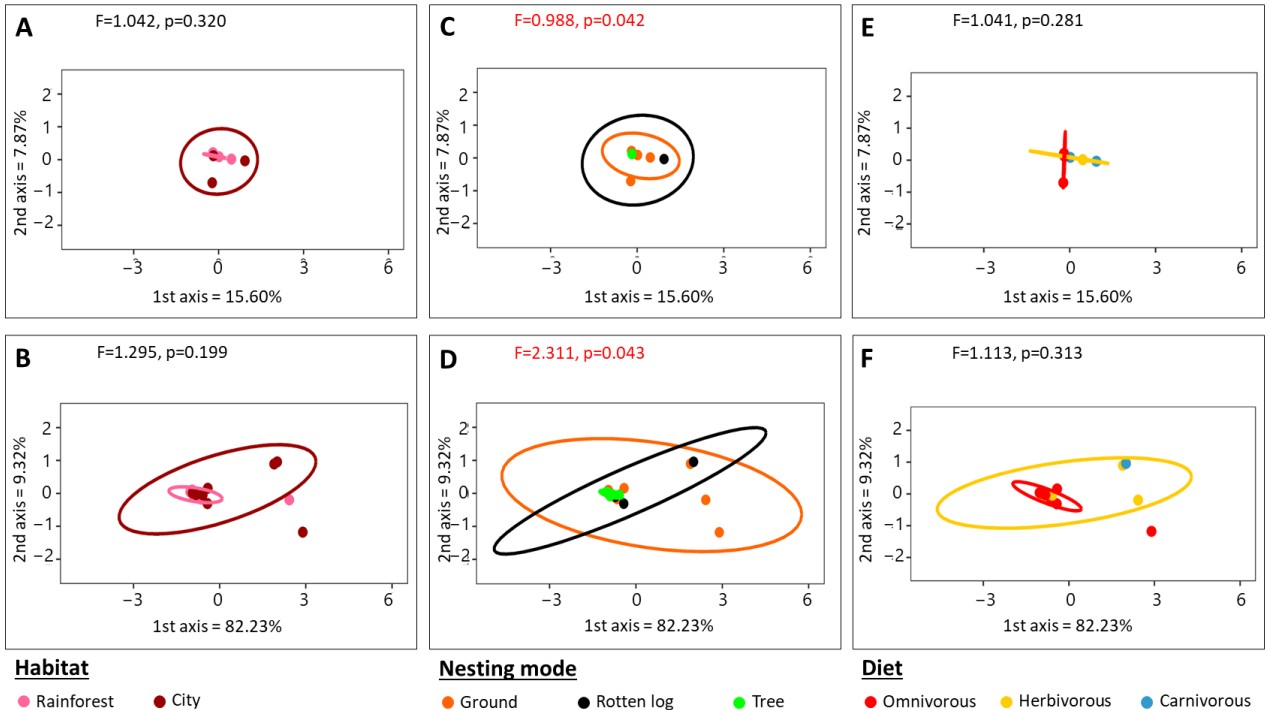

**Figure 4.** Differences in microbial eukaryote communities visualized with PCoAs across ant habitats (**A**,**B**), nesting modes (**C**,**D**), and diets (**E**,**F**). The distances used were the Bray–Curtis (**A**,**C**,**E**) and wUniFrac (**B**,**D**,**F**). The statistical *p*-values were obtained with pairwise PERMANOVAs. The circles represent confidence intervals of 0.95.

### 3.7. 18S rRNA ASV Similarity Percentage Analysis

To investigate which eukaryote subphyla are the most commonly associated with Amazonian ants, we used SIMPER analyses. The taxa bar plot of the total eukaryote relative abundance was largely dominated by undetermined eukaryotes for every factor tested (Figure 6A). Overall, besides the undetermined eukaryotes, the 10 most abundant eukaryote subphyla were Nematoda, Eugregarinorida, Chytridiomycota, Mortierellomycotina, Saccharomycotina, Ustilaginomycotina, Pezizimycotina, Basidiobolomycetes, Mucoromycotina, and Arthropoda. All statistics for the SIMPER analyses are presented in Supplementary File S7.

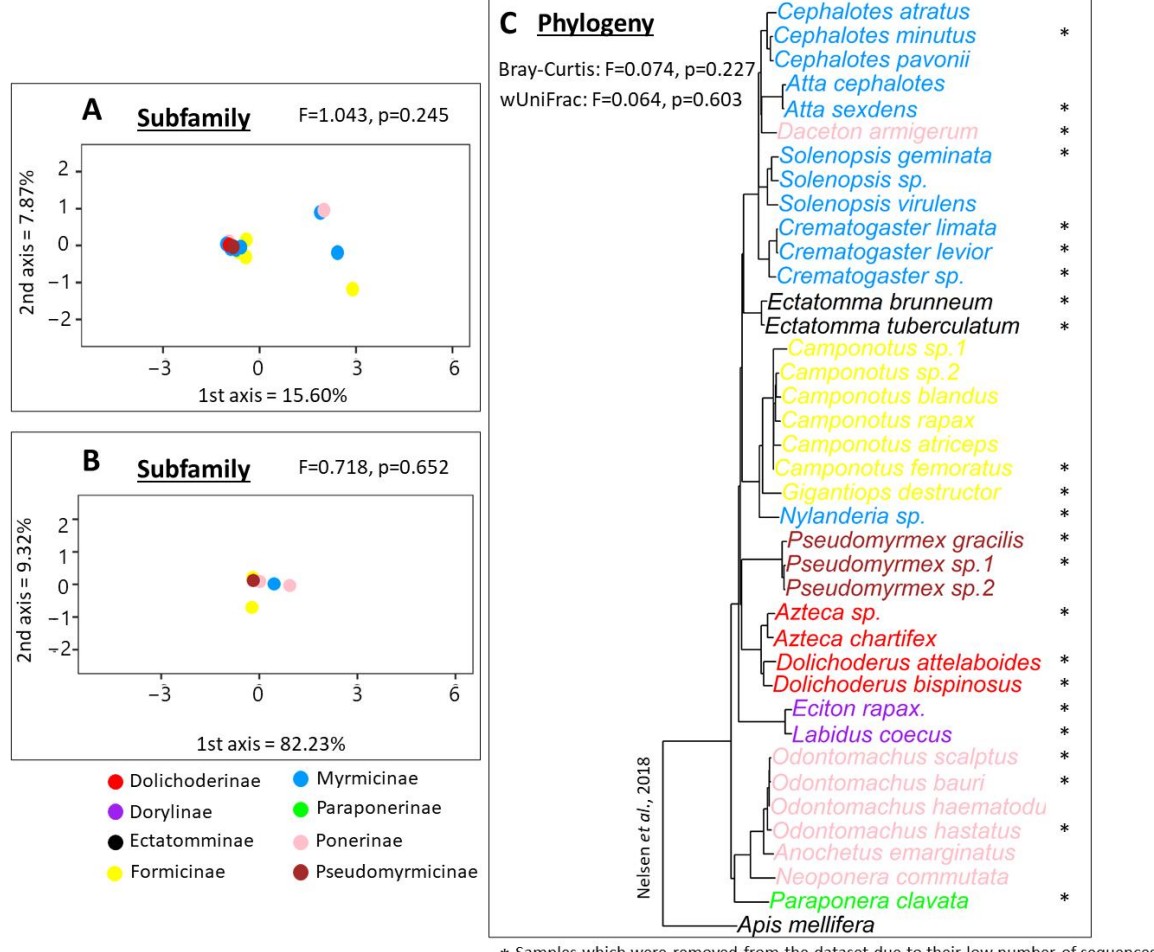

**Figure 5.** Differences in eukaryote communities visualized with PCoAs across ant subfamilies (**A**,**B**) and phylogenies (**C**). The distances used were the Bray–Curtis (**A**,**C**) and wUniFrac (**B**,**C**). The stars on the tips of the phylogenies denote samples that were excluded from this analysis due to low sequencing coverage. The statistical *p*-values were obtained with PERMANOVAs.

By focusing on the different ant habitats and nesting modes, undetermined eukaryotes were found to represent half of the total eukaryote relative abundance in city ants and ground nesting ants (49.10% and 61.90% respectively), while they represent almost 75% of the relative abundance in ants nesting in rotten logs and more than 98% in rainforest ants and ants nesting in trees. Nematoda are also very common in city ants and ground nesting or rotten log nesting ants (23.70%, 20.40%, and 23.40%, respectively). Eugragarinorida were present in city ants and in ground nesting ants (14.70% and 12.50%, respectively), while Chytridiomycota were only present in ground nesting ants.

The ant subfamilies Dolichoderinae and Pseudomyrmicinae were entirely associated with undetermined eukaryotes. Formicinae and Myrmicinae also contained mainly undetermined eukaryotes (72.70% and 73.90% respectively), but Formicinae also contained Eugragarinorida (19.60%), while Myrmicinae contained Nematoda and Chytridiomycota (13.50% and 11.30%, respectively). Ponerinae were composed of undetermined eukaryotes and Nematoda (51.40% and 48.60%, respectively).

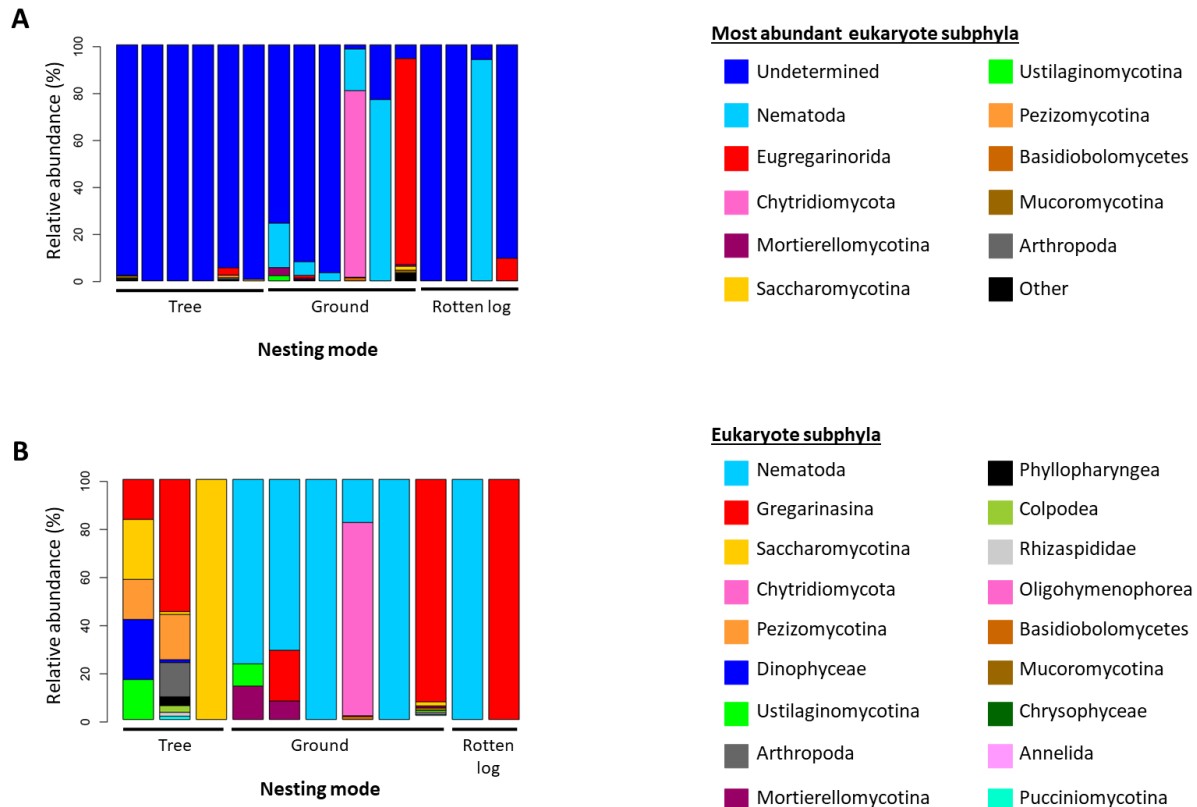

**Figure 6.** (**A**) Taxa bar plot based on the relative abundance ordered by ant nesting mode with all ASVs. The relative abundance bars are colored by the percentages of eukaryote subphyla. The 11 most abundant eukaryote subphyla associated with Amazonian ants are listed. The less abundant eukaryote subphyla are grouped under the term "Other". The term "Undetermined" represents eukaryote ASVs that could not be identified with the SILVA taxonomy. (**B**) Taxa bar plot based on the relative abundance ordered by ant nesting mode after removing all undetermined eukaryote ASVs. The relative abundance bars are colored by the percentages of eukaryote subphyla. A description of each eukaryote subphyla and examples of genera associated with each subphylum can be found in Supplementary File S11.

We used a SIMPER analysis to determine the contributions of the 11 most abundant eukaryote subphyla to the dissimilarities observed previously. All statistics obtained from the SIMPER analyses are presented in Supplementary File S9. There were no statistical differences in any eukaryote subphyla between ant habitats, diets, or subfamilies. However, there were statistical differences in some eukaryote subphyla between nesting modes. Indeed, undetermined eukaryotes had increased relative abundances in tree nesting ants compared with in ground nesting ants. In contrast, Nematoda had an increased relative abundance in ground nesting ants compared with in tree nesting ants (Figure S15). Pairwise Kruskal–Wallis comparisons for these two eukaryote subphyla are presented in Supplementary File S10.

Due to the high number of undetermined ASVs in our samples, we decided to remove these sequences to obtain a better representation of the identified eukaryote subphyla results (Figure 6B). The most abundant eukaryote subphyla were Nematoda and Gregarinasina. As shown before, Nematoda are more abundant in ground nesting ants than in tree nesting ants (Figure 6B). Most genera are associated with only a few different eukaryote subphyla, with one subphylum being predominant in the sample and the other subphyla having a low abundance. However, *Cephalotes* is the only genus in which the samples were associated with at least five different eukaryote subphyla with no subphylum being dominant, but instead, the different eukaryote subphyla were present with similar relative abundances. A

description of each eukaryote subphylum and examples of genera associated with each subphylum can be found in Supplementary File S11.

## 4. Discussion

### 4.1. Dietary Niche Structures Bacterial Communities but Not Microbial Eukaryote Communities Associated with Amazonian Ants

Our results show differences in bacterial diversity but not in bacterial relative abundance between carnivorous and herbivorous ants for both the Bray–Curtis and wUniFrac distances. Furthermore, our analyses highlight the finding that some bacterial orders are specific to ants with different diets. Indeed, Erysipelotrichales and Entomoplasmatales are more abundant in carnivorous ants, while Rhizobiales are more abundant in herbivorous ants. These results are in accordance with several previous studies that also highlighted the roles of these bacterial orders in complementing the host diet in ants [19,66,70] as well as in mammals [111]. Additionally, symbiotic bacteria have been shown to drive host evolution [68,112]. One of the possible explanations for this correlation between gut bacteria and the host diet could be the nutrient niche theory.

The structure of the gut bacterial community is hypothesized to be determined by the abundance and diversity of nutrients extracted from the host diet during digestion. The nutrient niche theory posits that gut ecological niches are determined by available nutrients in the gut [113–115]. This means that a specific bacteria species can only assert itself in the host gut if it is able to use a limiting nutrient. The nutrient niche theory has been supported by numerous diet supplementation studies, which have shown that the presence and abundance of specific bacterial species can be altered by experimentally modifying the types and abundance of nutrients present in the gut [116–120].

### 4.2. Microbial Community Structure Associated with Amazonian Ants Is Influenced by Abiotic Factors and Nesting Modes

Microbial community structure can be influenced by multiple abiotic factors, resulting in different habitat niches. Studies of abiotic factors in rainforests have shown that environmental parameters, like luminosity, humidity, and temperature, vary between the canopy and the forest floor [121,122]. Arthropods and free-living bacteria vertically structure their communities according to their different tolerance levels to these abiotic factors [123]. Some symbiotic bacteria can also confer to their host improved tolerance to these environmental perturbations, and this has previously been shown in several insects [124–128].

Our results indicated differences in microbial alpha and beta diversity, but not in bacterial abundance, between ground nesting ants and rotten log or tree nesting ants. Furthermore, as shown in our diet analysis, some bacterial orders are specific to a particular ant nesting mode. Indeed, Erysipelotrichales, Lactobacillales, and Rhizobiales are more abundant in ground nesting than in rotten log nesting ants, while Rickettsiales showed a higher abundance in rotten log nesting ants than in tree nesting ants. These bacteria could help their hosts to withstand specific environmental conditions, and future experimental work could demonstrate the roles of these bacteria.

The low numbers of currently available 18S rRNA sequences in public databases limit the study of microbial eukaryotes. In our study, the majority of the retrieved 18S rRNA sequences, after excluding any sequence identified as "Hymenoptera", could not be identified further than as being eukaryotes. Among the determined microbial eukaryote ASVs, the identification accuracy was variable with only a few ASVs assigned at the species level, while other AVSs could only be identified at the order or kingdom levels. This major issue has been discussed in previous work [129]. The fact that more undetermined microbial eukaryotes were found in tree nesting ants could come from the fact that, to date, there has been a greater number of studies on eukaryote diversity from soils [129–132], which could cause a lack of sequences from eukaryotes found primarily in the canopy. If this is the case, more studies on global eukaryote diversity from trees and arboreal species would be necessary to increase the eukaryote databases.

### 4.3. Bacterial Communities in Long-Term Association with Specific Ant Hosts Are Conserved

Several ant genera have well-established long-term relationships with specific bacterial communities. These bacterial communities are usually conserved within the different species of their host genus and often provide benefits to the host, as previously reported for *Camponotus* [19] and *Cephalotes* [70].

By focusing on the differences in bacterial beta diversity, we noticed that, overall, three genera were different from the other ant genera: *Camponotus* (subfamily: Formicinae), *Cephalotes* (subfamily: Myrmicinae), and *Odontomachus* (subfamily: Ponerinae). Looking deeper, we found that *Camponotus* are mainly composed of three bacterial orders: Enterobacterales (especially the bacterial genus *Blochmannia*), Acetobacterales, and Rickettsiales (especially the bacterial genus *Wolbachia*). *Blochmannia* are known to be the main symbiont of *Camponotus*, providing it with nutritional supplementation [133,134]. Some *Camponotus* species have also been shown to be strongly associated with Acetobacterales [135]. *Wolbachia* is a common insect symbiont and can have beneficial effects, like vitamin B supplementation in bedbugs [84,136], as well as negative effects when manipulating host reproduction [137]. Concerning the *Cephalotes* samples, we determined that they were mainly composed of four bacterial orders: Xanthomonadales, Burkholderiales, Rhizobiales, and Opitutales. Our results are in accordance with previous studies on *Cephalotes*, which highlighted that they possess a very stable core microbiome composed of five bacterial orders: Burkholderiales, Opitutales, Pseudomonadales, Rhizobiales and Xanthomonadales [66,69,112,138]. Together these symbionts synthesize amino acids via nitrogen recycling for their host [70]. Finally, focusing on the *Odontomachus* samples, our results show that they are mainly composed of *Rickettsiales* (especially the bacterial genus *Wolbachia*), *Rhizobiales*, and *Erysipelotrichales*. Few studies have focused on the bacterial diversity associated with *Odontomachus*, but a couple of studies have also shown strong associations of *Wolbachia* and Rhizobiales with *Odontomachus* ants [74,139]. To the best of our knowledge, the presence of Erysipelotrichales bacteria has not been previously studied in *Odontomachus* ants, but as it is a common symbiont of carnivorous species [140–142], its presence in predatory *Odontomachus* samples is not surprising.

In contrast to these very conserved microbiomes, we also identified two ant genera which displayed very high levels of species richness: *Ectatomma* and *Paraponera*. These two ant genera are characterized by not being associated with a core microbiome, but instead, have transient bacterial communities. Two studies have reported the bacterial communities associated with *Paraponera clavata* [73,143], but no clear identification of a core microbiome has been revealed, thus suggesting that the bacterial communities found are very variable and originate from the broad diets and habitats of these omnivorous ants.

### 4.4. Habitat Does Not Participate in Structuring Microbial Communities Associated with Amazonian Ants

As we have shown previously, microbial communities associated with ants are structured by host diet and nesting mode. Yet, these ecological factors may vary in different habitats. For example, ants may need to adapt their feeding habits, as different prey or plants may live in contrasting habitats. This is known as environmental filtering, a process in which the environment selects for and against certain species. Omnivorous ants or ants whose diet does not rely on a specific species could adapt faster and more easily to the resources they find in a new environment. Urbanization, in particular, is known to be associated with a variety of effects on arthropods, like pollution, habitat fragmentation, and a decrease in species richness [144,145]. Environment filtering has also been shown to affect microbial communities present in different habitats [146,147].

We did not find any correlations between the microbial communities from these ants in these contrasting habitats in term of diversity or abundance. Our results corroborate previous work investigating the differences in the microbial diversity of insects collected from different habitats that did not find any differences in the bacterial diversity between insects collected from urban environments and insects collected from rural environments [148,149].

We did find a statistical difference in the abundance of one bacterial order (Acetobacterales) between rainforest ants and city ants. However, this result represents an exception in our study, which suggests that environmental filtering does not affect the structure of ant microbial communities.

## 5. Conclusions

Our results show that microbial communities associated with Amazonian ants are structured by different factors. The bacterial communities associated with Amazonian ants are structured by the ant diet, nesting mode, and taxonomy, while the microbial eukaryote communities associated with Amazonian ants are only structured by the ant nesting mode. The ant habitat and evolutionary history were not shown to have any impact on structuring their associated microbial communities. Despite the large number of undetermined sequences of microbial eukaryotes, future work focusing on the co-occurrence between bacterial communities and microbial eukaryote communities could reveal microbe–microbe interaction dynamics inside the insect host. To the best of our knowledge, this is one of the first studies to focus on the microbial communities associated with a wide range of Neotropical ants. As such, future work on this topic would be useful to confirm our findings. In particular, since only one nest per species was collected in this study, research focusing on the microbiome of ants from several nests of the same species might increase the robustness of our findings.

**Supplementary Materials:** The following are available online at https://www.mdpi.com/xxx/s1, Figure S1: Sampling map; Figure S2: 16S rRNA rarefaction curves; Figure S3: 16S rRNA alpha diversity for ant habitat; Figure S4: 16S rRNA alpha diversity for ant nesting mode; Figure S5: 16S rRNA alpha diversity for ant diet; Figure S6: 16S rRNA alpha diversity for ant subfamily; Figure S7: 16S rRNA qPCR; Figure S8: 16S rRNA beta diversity for ants and nests; Figure S9: 16S rRNA SIMPER analysis; Figure S10: 18S rRNA rarefaction curves; Figure S11: 18S rRNA alpha diversity for ant habitat; Figure S12: 18S rRNA alpha diversity for ant subfamily; Figure S13: 18S rRNA alpha diversity for ant nesting mode; Figure S14: 18S rRNA alpha diversity for ant diet; Figure S15: 18S rRNA SIMPER analysis; Supplementary File S1: All samples; Supplementary File S2: 16S rRNA alpha diversity metrics; Supplementary File S3: Bacterial relative abundance; Supplementary File S4: 16S rRNA beta diversity metrics; Supplementary File S5: 16S rRNA SIMPER statistics; Supplementary File S6: 16S rRNA SIMPER pairwise comparisons; Supplementary File S7: 18S rRNA alpha diversity metrics; Supplementary File S8: 18S rRNA beta diversity metrics; Supplementary File S9: 18S rRNA SIMPER statistics; Supplementary File S10: 18S rRNA SIMPER pairwise comparisons; Supplementary File S11: Eukaryote phyla.

**Author Contributions:** Conceptualization, A.C., C.S.M. and C.D.; methodology, A.C., C.S.M. and C.D.; software, A.C.; formal analysis, A.C.; writing—original draft preparation, A.C.; writing—review and editing, A.C., C.S.M. and C.D.; funding acquisition, A.C., C.S.M. and C.D. All authors have read and agreed to the published version of the manuscript.

**Funding:** This research was funded by "Investissement d'Avenir", grant managed by Agence Nationale de la Recherche, CEBA, ref. ANR-10-LABX-25-01 to C.D., and the National Science Foundation, NSF DEB 1900357 to C.S.M. We are grateful to the University of French Guiana for A.C.'s Ph.D. fellowship and traveling grants.

**Informed Consent Statement:** Not applicable.

**Data Availability Statement:** All raw sequence data are publicly available NCBI SRA accession number PRJNA743571, and BioSample SAMN20055534 and SAMN20055535 respectively for the 16S rRNA and the 18S rRNA raw sequences.

**Acknowledgments:** The authors are grateful to Axel Touchard and Peter Flynn for their help with the collection and taxonomic identification of the ant samples. The authors are grateful to the Argonne National Laboratory (Lemont, IL, USA) for the 16S rRNA and 18S rRNA amplification and sequencing. The authors are thankful to the Cornell Institute of Biotechnology and the Cornell Information Technology support for their help in the setting up of the software used in this study. The authors are also thankful to Manuela Ramalho for all her assistance with statistical analysis.

**Conflicts of Interest:** The authors declare no conflict of interest and the funders had no role in the design of the study, in the collection, analyses, or interpretation of data, in the writing of the manuscript, or in the decision to publish the results.

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
