# Peer review of "Impact of Nesting Mode, Diet, and Taxonomy in Structuring the Associated Microbial Communities of Amazonian Ants"

_diversity, doi:10.3390/d15020126_

Round 1

Reviewer 1 Report

The research on the structure of bacterial and microbial eukaryote communities associated with insects is relevant and essential. Uncovering the diversity and function of insect-associated microbes contribute to our understanding of factors structuring the microbial communities associated with insects and their role. The current manuscript by Chanson et al. describes microbial communities associated with a wide range of ant Amazonian ant species, which were collected from different habitats and consumed different types of food and nesting habits. In contrast to some previous studies, they found that habitat does not structure the microbial communities of ants. However, ant diet and nesting mode impact bacterial communities, while only nesting mode structure microbial eukaryote communities. The manuscript is well written, and experiments are planned very well. I have a few comments in the attached file.

Author Response

Dear Editor,

We thank you and the two reviewers very much for the positive comments and valuable suggestions that helped us to improve our manuscript. You will find herewith a revised version accounting for referees’ comments and suggestions, as well as our replies below.

We provide our replies below starting by “>>>>” , to facilitate the reading. Again, thank you for your consideration.

Reviewer Comments

Reviewer #1:

1. L27. The introduction provides all necessary information but is very wordy and might be condensed slightly, notably the first two paragraphs.

>>> The first two paragraphs have been condensed.

2. L 109-118.

>>> There seems to be a comment on the last paragraph of the introduction because it is highlighted, but if there is a comment, we can’t read it.

3. L 122. Please mention how collected samples were identified up-to species level? More information about method of collection also needed here

>>> Samples were collected in March 2018. Ant samples were collected in the field using pincers and stored in small individual tubes containing EtOH, then tubes were put at -20°C. Nest samples were collected in the field and stored in sterile small bags, then bags were put at -20°C. Collected ants samples were identified up to the species level, when possible, in the field with a magnifying glass. Identifications were further refined in the lab with a binocular magnifier, using the key to subfamilies of the Neotropical region from Baccaro et al. (2015). These informations have been added in the methods section.

4. Figure 1. Please add legend on x-axis and its unit e.g. %.

>>> Legend and unit have been added.

5. Figure 6. Please label x-axis and write its unit for both A and B figs..

>>> Legend and unit have been added.

  1. Discussion. Remove figure numbers in the discussion.

>>> Changes have been made

7. L466. Add reasoning here with references.

>>> We do not think a reasoning should be added here because it is indicated in the last sentence at the end of the paragraph.

  1. L492-497. These are results not discussion.

>>> These lines have been removed from the discussion

9. L529. And or habitat.

>>> This line has been added

10. L543-547, and L558-564. Remove these lines.

>>> Changes have been made

Reviewer 2 Report

The authors collected 49 samples of ants (after supplementary file 1) representing 38 ant species, and 45 and nest samples. They conducted DNA extraction to explore microbial diversity to answer three different questions: 1) are correlations between microbial communities and the evolutionary history of their host? 2) Are different microbial communities associated to ant diet and nesting mode? 3) Are differences in the microbial communities between ants from the city and ants from the rainforest?

The manuscript is well written, with a plethora of data and results in both the main text and the supplementary files. The investigation of wild microbial communities associated to social insects is an important topic that the authors address with appropriate methods and statistical technics. I feel, however, that the manuscript needs extra work or could be split into different works to gain clarity. The main reason is based on the nature of the samples, that drive to different questions: on the one hand, they investigate the microbial community of ants, and on the other, nest samples. By analysing single ant microbial communities, you will analyse and compare, overall, the gut and body microbiome across species, which is somehow correlated with the genetic background and physiology or diet of the host; and, although there exist some previous works (Atta, Cephalotes, Daceton…), those authors focus mainly on single species. Then, by analysing the nest, you will provide insight into the potential role of the ecology and behaviour of the ants by selecting communities. Nevertheless, if you decide to keep it as a whole, I strongly recommend to clearly differentiate between ant and nest analysis in the results.

Major concerns

1) I am worried about the lack of replicates both at the ant and the nest species levels: Rarefaction curves work well to assess that you have enough reads within each individual ant DNAs, but how do you justify using just one individual nest per species (e.g. are the communities of both Dolichoderus species from the rainforest naturally different or by chance because of the sampling?). Don’t misunderstand me, your results are wealthy, but not robust. You might indicate this in conclusions (e.g L570).

2) Why don’t you analyse together all the microbial community (bacteria and eukaryote for ants and nests separately)?

3) I would appreciate the analysis at the microbial genus level

Minor concerns:

1) I think pertinent to add some new references in the introduction because you weight too much eukaryotic parasites and the following describe mutualist or non-pathogenic associations (description of fungi present in ant nests for structural or hunting purposes):

Defossez et al. 2009. New Phytol 182

Ruiz-Gonzalez et al. 2011. Biol Lett 7

Nepel et al. 2014. PLoS One 9.

2) L123-130: When were the samples collected?

- How were the nests collected?

- Describe the protocols and the aseptic measures to avoid contamination.

3) In supplementary information, Fig S1: describe the ecological differences across sampling places within the Nouragues (the four sites) and within the urban areas (e.g. urban parks).

- The pictures do not have enough quality and get blurred.

3) In supplementary file 1, you might want to add a column displaying the site of collection at Nouragues or the city (site i-viii).

- When you say savana, do you mean savannah?

- Any chance to know the plant host species for the host trees?

- I suggest ordering the table by ant/nest (as it is now) and then by ant subfamily, genus and species. Thus, it will mirror Fig 1.

- Sample type column is not informative, delete it.

- Which is the difference between “habitat mode” and “habitat”?

4) in L150-159 and Supplementary file 1: the linker primer sequences are the same for all samples, why don’t you write down this in the legend, and then save columns for extra clarity (16S and 18S)?

- Both the 16S barcode and linker primer sequences are the same than the 18S sequences: a mistake?

- Are 515F and F04 sequences the same for the linker primer sequences in supp file 1 (they are not)?

5) L219-220: Did the data meet the ANOVA’s assumptions?

6) L239: avoid the use of side-by-side parentheses.

7) Here you indicate the same sample size as in the supplementary file 1. Maybe it is wrong in L123-125?

8) Supplementary file 2: Which criteria (cut off) to decide the abundance of an order? The table has a weird pattern of line thickness that does not correspond to each category.

9) L270-271: In supplementary file 3 the information is about ant subfamily, not genus.

10) Results in main text and supplementary information: provide statistics in italics, please.

11) L289-292: as expected, there are differences when comparing gut microbiome and nest microbiome. This should be the starting point of the results. Then, it is natural to analyse ants and nest separately.

12) L298: Fig S4. provide degrees of freedom for each t-test and ANOVA test.

13) Figs 2 – 5. Same scale for axis. Smaller and uniform size dots might make clearer the graphs. With the legend on the bottom of each pair of figures you might shrink the figure height and save publishing space.

14) Figs 3C and 5C. It might help to colour the members of each subfamily after the code in fig 3A, B. As in previous comment, if only one legend of colours is set under both fig 3A and B, you save space.

- L325-328, 369-371. Are the Mantel tests conducted on nest microbiome or ant microbiome? The latter might be more conservative across phylogeny if it is vertically transmitted…

15) Supplementary file 4 is lacking. You do not cite supplementary files 5 and 7 in main text

16) Fig S5. Same scale across axes. Indicate the meaning of letters in graphs.

17) L382-403. Again, I do not know whether you are providing overall joint ant and nest results, one or the other. I think more informative to analyse individuals (16S and 18S together) and nests (16S and 18S together).

18) L411: Instead of examples, I would love to see the whole list of genera or species that you found.

- In supplementary information pages 30 and 31, I think that “mean read number per group” should not be expressed in (%).

19) Supplementary file 9: “influential”.

20) Figure 6 A. Why are 16 bars?

- Fig 6 B. Gregarinasina (Subclass) or Eugregarinorida (Order)?

- Fig 6 B. It might be more informative to write above each column the % of taxa represented before removing the undetermined taxa and the total number of genera.

21) L444: Rhizobiales has been found to be very well represented in the carnivorous Allomerus ants. So I suggest: “…ants[19,62,66,106]” where 106 ref should be Ruiz-Gonzalez et al. 2019. Insects10. A potential explanation might be that their living in close association in a three-way symbiosis with a tree host and a fungus that transfer nitrogen to the plant, might select for bacteria that boost nitrogen fixation for the plant after obtaining it from the preys they hunt.

22) Fig S8, S9, S10, S11, S12, S13, S14, S15 are lacking.

23) L500: avoid side-by-side parentheses.

24) L501-508. If you analyse the diversity at the level of genus, you can find the distribution of Blochmannia and Wolbachia symbionts across ant species.

25) L528: “…, thus suggesting…” instead of “…suggested…”

26) L559: Add “…habitat, geography, and evolutionary history.”

27) Check all references. There are many typos and some do not follow the ACS style. Always capitalize genus names but never species names, and write them in italics.

Author Response

Dear Editor,

We thank you and the two reviewers very much for the positive comments and valuable suggestions that helped us to improve our manuscript. You will find herewith a revised version accounting for referees’ comments and suggestions, as well as our replies below.

We provide our replies below starting by “>>>>” , to facilitate the reading. Again, thank you for your consideration.

Reviewer Comments

Reviewer #2:
Major concerns:

1. I am worried about the lack of replicates both at the ant and the nest species levels: Rarefaction curves work well to assess that you have enough reads within each individual ant DNAs, but how do you justify using just one individual nest per species (e.g. are the communities of both Dolichoderus species from the rainforest naturally different or by chance because of the sampling?). Don’t misunderstand me, your results are wealthy, but not robust. You might indicate this in conclusions (e.g L570).

>>> This point was added in the conclusion

2. Why don’t you analyse together all the microbial community (bacteria and eukaryote for ants and nests separately)?

>>> We could not analyze together all the microbial community, because for the microbial eukaryote dataset 33 ant samples out of the 49 tested did not reach the minimum sampling depth of 199 reads and had to be excluded from the analysis.

3. I would appreciate the analysis at the microbial genus level.

>>> An analysis at the bacterial genus level is difficult since some of the most abundant taxa can only be identified up to the family level. It is possible to resume the bacterial genera present in some of the samples in a figure similar to Figure 1, but we believe it would be more descriptive than analytic since numerous bacterial genera will remain undetermined.

Minor concerns:

1) I think pertinent to add some new references in the introduction because you weight too much eukaryotic parasites and the following describe mutualist or non-pathogenic associations (description of fungi present in ant nests for structural or hunting purposes): Defossez et al. 2009. New Phytol 182 ; Ruiz-Gonzalez et al. 2011. Biol Lett 7 ; Nepel et al. 2014. PLoS One 9.

>>> These references have been added in the introduction.

2) L123-130: When were the samples collected? How were the nests collected? Describe the protocols and the aseptic measures to avoid contamination.

>>> Samples were collected in March 2018. Ant samples were collected in the field using pincers and stored in small individual tubes containing EtOH, then tubes were put at -20°C. Nest samples were collected in the field and stored in sterile small bags, then bags were put at -20°C. Collected ants samples were identified up to the species level, when possible, in the field with a magnifying glass. Identifications were further refined in the lab with a binocular magnifier, using the key to subfamilies of the Neotropical region from Baccaro et al. (2015). These informations have been added in the methods.

3) In supplementary information, Fig S1: describe the ecological differences across sampling places within the Nouragues (the four sites) and within the urban areas (e.g. urban parks). The pictures do not have enough quality and get blurred.

>>> Unfortunately we can’t have a better quality for this picture

4) In supplementary file 1, you might want to add a column displaying the site of collection at Nouragues or the city (site i-viii). When you say savana, do you mean savannah? Any chance to know the plant host species for the host trees? I suggest ordering the table by ant/nest (as it is now) and then by ant subfamily, genus and species. Thus, it will mirror Fig 1. Sample type column is not informative, delete it. Which is the difference between “habitat mode” and “habitat”?

>>> We removed a few columns, including the habitat and sample type column. The column habitat mode is already displaying the site of collection for each sample between Nouragues (Rainforest) and the city. We are not able to identify the host trees. The table has been reordered as suggested.

5) in L150-159 and Supplementary file 1: the linker primer sequences are the same for all samples, why don’t you write down this in the legend, and then save columns for extra clarity (16S and 18S)? Both the 16S barcode and linker primer sequences are the same than the 18S sequences: a mistake? Are 515F and F04 sequences the same for the linker primer sequences in supp file 1 (they are not)?

>>> The linker primer sequences were amended and indicated in the legend of the table as suggested.

6) L219-220: Did the data meet the ANOVA’s assumptions?

>>> Yes the data met the ANOVA’s assumptions (normality and homogeneity of variances).

7) L239: avoid the use of side-by-side parentheses.

>>> One set of parentheses was removed.

8) Here you indicate the same sample size as in the supplementary file 1. Maybe it is wrong in L123-125?

>>> We reformulated this part of the method to make it more understandable.

9) Supplementary file 2: Which criteria (cut off) to decide the abundance of an order? The table has a weird pattern of line thickness that does not correspond to each category.

>>> I believe the reviewer meant Supplementary file 3. We choose to focus on the 15 most abundant bacterial orders, because they were present in at least 10% in one of the samples. We corrected the line thickness in the table.

10) L270-271: In supplementary file 3 the information is about ant subfamily, not genus.

>>> The line was amended.

11) Results in main text and supplementary information: provide statistics in italics, please.

>>> Statistics have been put in italics

12) L289-292: as expected, there are differences when comparing gut microbiome and nest microbiome. This should be the starting point of the results. Then, it is natural to analyse ants and nest separately.

>>> We add a sentence to help with the flow.

13) L298: Fig S4. provide degrees of freedom for each t-test and ANOVA test.

>>> Degrees of freedom were added

14) Figs 2 – 5. Same scale for axis. Smaller and uniform size dots might make clearer the graphs. With the legend on the bottom of each pair of figures you might shrink the figure height and save publishing space.

>>> Figures were modified as suggested

15) Figs 3C and 5C. It might help to colour the members of each subfamily after the code in fig 3A, B. As in previous comment, if only one legend of colours is set under both fig 3A and B, you save space. L325-328, 369-371. Are the Mantel tests conducted on nest microbiome or ant microbiome? The latter might be more conservative across phylogeny if it is vertically transmitted…

>>> Figures were modified as suggested. The Mantel tests are conducted on ant microbiome.

16) Supplementary file 4 is lacking. You do not cite supplementary files 5 and 7 in main text

>>> Supplementary file 4 is in the SI. Supplementary files 5 and 7 were added in the main text

17) Fig S5. Same scale across axes. Indicate the meaning of letters in graphs.

>>> Figure was modified as suggested.

18) L382-403. Again, I do not know whether you are providing overall joint ant and nest results, one or the other. I think more informative to analyse individuals (16S and 18S together) and nests (16S and 18S together).

>>> We are analyzing the ant microbiome. We only looked at the nest microbiome to ensure that it was different than the ant microbiome, and so that it could be investigated on its own. As you pointed out earlier, the study of the nest microbiome would be an interesting but separate analysis, which was not the goal of this research.

19) L411: Instead of examples, I would love to see the whole list of genera or species that you found. In supplementary information pages 30 and 31, I think that “mean read number per group” should not be expressed in (%).

>>> As most of the taxa could not be identified further than the subphyla level, we can’t provide a whole list of eukaryote genera present in the ant microbiome. Therefore instead we chose to list some examples of genera identified in the ant microbiome and known to be associated with insects, soil or plants. In supplementary information pages 30 and 31, the mistakes were amended.

20) Supplementary file 9: “influential”.

>>> The term was amended

21) Figure 6 A. Why are 16 bars? Fig 6 B. Gregarinasina (Subclass) or Eugregarinorida (Order)? Fig 6 B. It might be more informative to write above each column the % of taxa represented before removing the undetermined taxa and the total number of genera.

>>> There are 16 bars, because 33 ant samples out of the 49 tested did not reach the minimum sampling depth of 199 reads and had to be excluded from the analysis.

22) L444: Rhizobiales has been found to be very well represented in the carnivorous Allomerus ants. So I suggest: “…ants[19,62,66,106]” where 106 ref should be Ruiz-Gonzalez et al. 2019. Insects10. A potential explanation might be that their living in close association in a three-way symbiosis with a tree host and a fungus that transfer nitrogen to the plant, might select for bacteria that boost nitrogen fixation for the plant after obtaining it from the preys they hunt.

>>> It is indeed a possible explanation, but it does not seem the right place to add this very specific comment in the introductory paragraph of the discussion.

23) Fig S8, S9, S10, S11, S12, S13, S14, S15 are lacking.

>>> Fig S8 to S15 are in the SI.

24) L500: avoid side-by-side parentheses.

>>> The second set of parentheses was removed.

25) L501-508. If you analyse the diversity at the level of genus, you can find the distribution of Blochmannia and Wolbachia symbionts across ant species.

>>> As stated earlier, an analysis at the bacterial genus level is difficult since some of the most abundant taxa can only be identified up to the family level.

26) L528: “…, thus suggesting…” instead of “…suggested…”

>>> The line was amended.

27) L559: Add “…habitat, geography, and evolutionary history.”

>>> This part of the conclusion was removed.

28) Check all references. There are many typos and some do not follow the ACS style. Always capitalize genus names but never species names, and write them in italics.

>>> The references were amended when needed.
